# Beneficial effects of choir singing on cognition and well-being of older adults: Evidence from a cross-sectional study

Emmi Pentikäinen[1]*, Anni Pitkäniemi[1], Sini-Tuuli Siponkoski[1], Maarit Jansson[1], Jukka Louhivuori[2], Julene K. Johnson[3], Teemu Paajanen[4], Teppo Särkämö[1]

1 Cognitive Brain Research Unit, Department of Psychology and Logopedics, Faculty of Medicine, University of Helsinki, Helsinki, Finland, 2 Department of Music, Art and Culture Studies, University of Jyväskylä, Jyväskylä, Finland, 3 Institute for Health & Aging, University of California San Francisco, San Francisco, California, United States of America, 4 Finnish Institute of Occupational Health, Helsinki, Finland

* emmi.pentikainen@helsinki.fi

**Data Availability Statement:** The data file is available from OSF repository (https://osf.io/fwnr3/).

## Abstract

### Background and objectives

Choir singing has been associated with better mood and quality of life (QOL) in healthy older adults, but little is known about its potential cognitive benefits in aging. In this study, our aim was to compare the subjective (self-reported) and objective (test-based) cognitive functioning of senior choir singers and matched control subjects, coupled with assessment of mood, QOL, and social functioning.

### Research design and methods

We performed a cross-sectional questionnaire study in 162 healthy older (age ≥ 60 years) adults (106 choir singers, 56 controls), including measures of cognition, mood, social engagement, QOL, and role of music in daily life. The choir singers were divided to low (1–10 years, N = 58) and high (>10 years, N = 48) activity groups based on years of choir singing experience throughout their life span. A subcohort of 74 participants (39 choir singers, 35 controls) were assessed also with a neuropsychological testing battery.

### Results

In the neuropsychological testing, choir singers performed better than controls on the verbal flexibility domain of executive function, but not on other cognitive domains. In questionnaires, high activity choir singers showed better social integration than controls and low activity choir singers. In contrast, low activity choir singers had better general health than controls and high activity choir singers.

**Funding:** TS Academy of Finland: grants 299044, 305264, 306625 https://www.aka.fi/en/ European Research Council: grant 803466 https://erc.europa.eu/ The funders had no role in study design, data collection and analysis, decision to publish, or preparation of the manuscript.

**Competing interests:** The authors have declared that no competing interests exist.

## Discussion and implications

In healthy older adults, regular choir singing is associated with better verbal flexibility. Long-standing choir activity is linked to better social engagement and more recently commenced choir activity to better general health.

## Introduction

As the population ages and the proportion of older adults grows, it has become increasingly important to find new ways to support and improve their quality of life and well-being. Not only age-related neurodegenerative diseases, but also normal aging induces many changes in brain function and cognition as well as in social and physical conditions of life that can significantly affect individual's well-being. Cognitively, healthy aging is known to affect primarily abilities that require flexible recruitment and modification of task-appropriate skills, while general knowledge acquired earlier in life is preserved [1]. Processing speed and executive functions, such as inhibitory control and selective attention, seem to be vulnerable to age-related changes in the brain [2]. These changes in cognition can be linked to atrophy in specific brain areas, including the prefrontal cortex, occurring with normal aging [3]. Cognitive decline is often linked also to emotional deficits, and, for example, depression is a common reactive symptom in the early stages of dementia [4]. On the other hand, depression has been identified as an important risk factor for cognitive decline and dementia [5]. Socially, aging is also associated with reduced social networks and increased loneliness, which can seriously impair both cognitive and emotional well-being as well as physical health [6, 7]. Overall, as cognitive, emotional, and social well-being are so closely intertwined, it is of great importance to find new versatile ways to enhance the general well-being of older adults.

Although age-related changes occur, the brain remains plastic throughout life and ageing is not associated only with negative neural changes: the aging brain can adapt structurally and functionally to compensate for lost volume and functioning of specific areas, for example by recruiting frontal areas more extensively and reducing bilateral asymmetry in carrying out functions [8]. Evidence from intervention studies suggests that cognitive training can have positive effects on cognition and brain structure and function in older adults [8, 9]. Similarly, research on cognitive reserve indicates that there are a number of protective lifestyle-related factors, such as education and cognitive, social, and physical activities, that can help preserve cognitive functioning even when neurodegenerative changes are already apparent [5]. There is now growing interest towards everyday activities that can help preserve cognitive function in old age. Physical exercise is perhaps the most studied of these, and there is mounting evidence that it can decrease the risk of cognitive decline and dementia as well as slow down age-related neural changes [10]. Recently, attention has turned also to the role of cultural activities and arts, especially those involving music, as effective tools to improve health and well-being, in terms of prevention and management of illnesses [11] and in the context of aging [12].

### Musical activities and healthy aging

Converging findings from psychological, clinical, and neuroimaging research suggest that music is an important source of enjoyment, learning, and well-being as well as a versatile stimulus for the brain, also during aging [13]. Music is commonly used to regulate mood and arousal, communicate and interact with others, and enrich everyday life. This can be observed also in times of crisis; for example, during the COVID-19 pandemic, people quarantined to their homes turned to playing music and singing together from their open windows and

balconies to help cope with emotional distress and social isolation. In the brain, music engages multiple cognitive, motor, emotional, and social processes mediated by a wide-scale, largely bilateral network of cortical and subcortical brain regions [14, 15]. Musical training has been found to enhance cognitive performance, with transfer effects on executive functions, attention, and memory [16, 17], and induce structural and functional neuroplasticity changes [18], especially in temporal, frontal, parietal, and cerebellar regions associated with higher-level auditory-cognitive functions [19]. Notably, some evidence suggests that prefrontal cortical regions, which are most vulnerable to age-related atrophy, are more preserved in older musicians than in older control subjects [20] and that music making has an age-decelerating effect on brain structure [21]. Lifetime instrumental musical activities and training to play a musical instrument at old age have been linked to better cognitive flexibility, processing speed, working memory, and verbal and non-verbal memory [22–25].

## Choir singing and healthy aging

While the neurocognitive impact of instrumental musical training has been extensively studied, little is known about the potential effects of singing on neuroplasticity and cognitive functioning during aging. For the brain, singing is a highly versatile and multi-domain process, requiring the complex interplay of auditory, vocal-motor, linguistic, cognitive, and emotional processes. Neuroimaging studies suggest that singing entails continuous interaction between two cortical systems, the parietal-frontal (dorsal) vocal production pathway and the temporal-frontal (ventral) auditory perception pathway, which work together as a loop to enable fine vocal motor control based on somatosensory and auditory feedback [26]. In addition to these core systems, also other prefrontal, limbic, and cerebellar areas linked to attention, working memory, rhythm, and emotion are engaged during singing perception and production [27–29].

Among different musical activities, choir singing is the most popular and widespread hobby, also among seniors. In Europe, there are 37 million choir singers, and participation in senior choirs is growing rapidly [30]. The coupling of singing-related brain processes (vocal-motor, auditory, linguistic, cognitive, emotional) with the social interaction (singing together in a group) and goal-directed learning (learning to sing and perform polyphonic song arrangements) elements makes choir singing a particularly promising activity for promoting cognitive reserve and psychological and social well-being in aging. Previous research on group singing has shown that it can improve mental health and emotional and social well-being in adults who have a mental health condition [31]. Physiologically, singing has a positive impact on cardiorespiratory functions [32, 33], and the emotional gains of singing are linked to the secretion of endocannabinoids, immunoglobulins, and cortisol [34, 35]. In older adults, regular participation in community-level choirs can reduce anxiety, depression, and loneliness; improve self-evaluated quality of life (QOL), physical health, and interest in life; and increase general activity [36–39].

## Potential cognitive benefits of choir singing in older adults

Evidence for the potential cognitive benefits of choir singing in healthy older adults is scarce, thus far limited to a single study. In a pilot study of healthy older adults (N = 49), Fu et al. [33] reported improved performance in verbal fluency and memory tests after a 12-week group singing program, but lack of a control group limits the conclusions that can be drawn from this result. No longitudinal study has yet been conducted to explore the long-term effects of choir singing in the elderly. Here, we report baseline results of an ongoing longitudinal study in Helsinki where a cohort of elderly choir singers and non-singer control subjects are

followed over a three-year period using questionnaires, neuropsychological tests, and electro-encephalography (EEG) measurements. In this cross-sectional comparison of active choir singers and demographically matched controls (total N = 162), our hypotheses were that choir singers show better cognitive performance, especially in tasks measuring executive functions, as well as better self-reported mood, social well-being and QOL compared to the controls. Moreover, we sought to explore whether the length of the singing experience affected the potential benefits of choir singing.

## Design and methods

### Participants and study design

The study was approved by the Ethical Review Board in the Humanities and Social and Behavioural Science of the University of Helsinki. All participants gave written informed consent. The participants were 162 older adults recruited from the Adult Education Centers of the Cities of Helsinki, Espoo and Vantaa and from different senior citizens' associations and independent choirs in the Helsinki area. Participants were recruited through presentations, flyers, and e-mail advertisements. The cross-sectional study reported here is from the baseline data of an ongoing longitudinal cohort study of the effects of senior choir singing on neurocognitive ageing. The present study comprises two parts: a main cohort study with questionnaire data from the full participant sample (N = 162) and a subcohort study with neuropsychological test data from a subsample of the participants (N = 74). The inclusion criteria in the main cohort study were (i) age ≥ 60 years, (ii) Finnish-speaking, and (iii) absence of neurological (e.g., dementia, stroke) or psychiatric (e.g., schizophrenia, bipolar disorder) disorders. In the subcohort study, additional inclusion criteria were (iv) absence of medication affecting CNS function, (v) absence of significant hearing loss, and (vi) absence of severe sleep disorder (e.g., insomnia, sleep apnea).

Of the 162 participants, 106 were choir singers (persons who currently sing in a choir and who had been singing for at least one year) and 56 were control subjects (persons who do not sing in a choir currently and have not participated in choir singing during the last 10 years). Choir singing was defined as regular participation in a choir, which (i) is led by a professional choir conductor, (ii) trains together at least once a week, and (iii) performs regularly (at least twice a year). For the main cohort study, the choir singers were divided to those who had started choir singing earlier in life and had sung in a choir for more than 10 years (referred to hereafter as "high activity choir singers") and those who had started choir singing later in life and had sung in a choir for 10 years or less (hereafter "low activity choir singers"). A similar classification was used previously by Hanna-Pladdy and MacKay [24] in their study of older adult musicians. This resulted in three relatively balanced groups for the main cohort study: (1) high activity choir singers (N = 48), (2) low activity choir singers (N = 58), and (3) control subjects (N = 56). In the subcohort study, the participants were choir singers (N = 39) and control subjects (N = 35).

### Neuropsychological testing

Neuropsychological testing was conducted for a randomly selected subsample (N = 74) of the main cohort. The testing (duration 1.5 h) was performed by a trained psychologist in a quiet testing room. The testing battery (see Table 1 for details) covered six cognitive domains: general cognition, executive functions, processing speed, working memory, episodic memory, and verbal skills. General cognition was evaluated with the Montreal Cognitive Assessment (MoCA) [40]. Executive functions were further divided to three subdomains: (i) verbal flexibility assessed with the Phonemic fluency test [41]; (ii) shifting assessed with a computerized (tablet) modification of the Trail Making Test, which is included in the Flexible Attention Test

**Table 1. Description of outcome measures.**

| Type | Domain | Measure | Description | Variables |
|------|--------|---------|-------------|-----------|
| Q | Cognitive function | CFQ | 25 items measuring cognitive failures in different everyday situations involving perception, attention, memory and motor functions | total score (sum) |
| | | PRMQ | 16 items measuring prospective and retrospective memory | total score (sum) |
| Q | Depression | CES-D | 20 items measuring depressive symptoms | total score (sum) |
| Q | Social well-being | SPS | 24 items measuring level of support from social relationships | total score (sum) |
| | | | 6 scales: Attachment, Social Integration, Reassurance of Worth, Reliable Alliance, Guidance, and Opportunity of Nurturance | 6 scale scores (sum) |
| Q | QOL | WHOQOL-Bref | 26 items measuring different aspects of quality of life | |
| | | | 4 scales: Physical, Psychological, Social, and Environmental QOL | 4 scale scores (sum) |
| | | | 2 separate items: Overall QOL and General Health | 2 item scores |
| Q | Role of music | MusEQ | 35 items measuring the use of music and its role in everyday life | total score (weighted) |
| T | General cognition | MoCA | 6 short tasks measuring visuospatial functions, verbal abilities, memory, attention and orientation | total score (sum) |
| T | EF: Verbal Flexibility | Phonemic fluency | List verbally as many words as possible during 60 seconds starting with the letter S | score |
| T | EF: Shifting | FAT | Trail Making Test: Connect numbers (Part A) and numbers and letters (Part B) as fast as possible | time difference (B-A) |
| T | EF: Inhibition | Simon task | Respond to red/blue square appearing on the left/right side of screen by pressing a button [congruent (CON) and incongruent (INC) trials] | time difference (INC-CON) |
| T | Processing speed | WAIS-IV | Visual Search: Copy symbols corresponding to numbers (2 min) | sum of raw scores |
| | | | Coding: Search rows of symbols for target symbols (2 min) | |
| T | Working memory | WAIS-IV | Digit span. Recall lists of numbers in different order | sum of raw scores |
| | | FAT | Visual span: Recall visuospatial patterns | |
| T | Problem solving | WAIS-IV | Arithmetic: Solve verbally presented arithmetic tasks | raw score |
| T | EM: Immediate | WMS-III | Logical memory I: Recall a story immediately after hearing it | sum of raw scores |
| | | | Word lists I: Recall a list of words immediately after hearing it | |
| T | EM: Delayed | WMS-III | Logical memory II: Recall a story after 30 min delay | sum of raw scores |
| | | | Word lists II: Recall a list of words after 30 min delay | |
| T | Verbal skills | WAIS-IV | Vocabulary: Explain the meaning of words | sum of raw scores |
| | | Semantic fluency | List verbally as many animals as possible during 60 seconds | |

Abbreviations: CES-D = Center for Epidemiologic Studies Depression scale, CFQ = Cognitive Failures Questionnaire, EF = executive function, EM = episodic memory, FAT = Flexible Attention Test, MoCA = Montreal Cognitive Assessment, MusEQ = Music Engagement Questionnaire, PRMQ = Prospective and Retrospective Memory Questionnaire, Q = questionnaire, SPS = Social Provisions Scale, T = neuropsychological test, WAIS-IV = Wechsler Adult Intelligence Scale IV, WHOQOL-Bref = Quality of Life Questionnaire of the World Health Organizations, WMS-III = Wechsler Memory Scale III.

(FAT) developed at the Finnish Institute of Occupational Health; and (iii) inhibition assessed with a tablet version of the Simon task [42]. Processing speed was evaluated with the Symbol search and Coding subtests of the Wechsler Adult Intelligence Scale IV (WAIS-IV) [43]. Working memory was assessed with the Digit span subtest of WAIS-IV and a tablet version of the Corsi Block-tapping test [44]. The Arithmetic subtest of WAIS-IV was used as a separate measure of problem solving. Episodic memory was evaluated using the Logical memory and Word lists subtests of the Wechsler Memory Scale III (WMS-III) [45]. Verbal skills were evaluated with the Vocabulary subtest from WAIS-IV and with the Semantic fluency task [41]. For each domain, sum scores of the individual tests were used in the analyses.

## Questionnaires

In the main cohort study (N = 162), six self-report questionnaires (see Table 1 for details) were used to measure cognitive functioning, depression, social well-being, QOL, and role of music

in daily life. Cognitive functioning was assessed with the Cognitive Failures Questionnaire (CFQ) [46] and the Prospective and Retrospective Memory Questionnaire (PRMQ) [47]. Depression was evaluated using the Center for Epidemiologic Studies Depression scale (CES-D) [48]. Social well-being was assessed with the Social Provisions Scale (SPS) [49]. QOL was assessed with the Quality of Life Questionnaire of the World Health Organizations (WHOQOL-Bref) [50]. Role of music in daily life was evaluated with the Music Engagement Questionnaire (MusEQ) [51].

As a background variable, level of cognitive and physical activity was assessed with six items from the Lifetime of Experiences Questionnaire (LEQ) [52], three concerning cognitive activity (non-musical arts, reading, language learning) and three concerning physical activity (mild, moderate and vigorous exercise) during old adulthood (from age 60 years). Also the frequency of group singing during young adulthood, middle age, and old adulthood was measured with an additional item of the LEQ questionnaire.

## Statistical analyses

Statistical analyses were performed with SPSS (IBM SPSS Statistics 25). Group differences in demographic background variables were analyzed with independent-samples t tests, one-way ANOVAs, and chi-square tests. Group differences in the outcome measures (questionnaires and neuropsychological tests) were analyzed with univariate ANCOVAs where those background variables, which showed group differences, were included as covariates. Post hoc tests were performed using the least significant difference (LSD) test.

## Results

### Demographic characteristics of the participants

The demographic characteristics of the participants are shown in Table 2. In the main cohort study, there were significant or marginally significant differences between the three groups in age [$F(2,159) = 5.76$, $p = 0.004$], gender [$\chi^2 (2) = 6.74$, $p = 0.034$], and living situation [$\chi^2 (2) = 5.76$, $p = 0.056$]. The high activity choir singers were on average older than the low activity choir singers ($p = 0.001$) and the controls ($p = 0.019$). The low activity choir singers had smaller proportion of women ($p = 0.012$) and persons living alone ($p = 0.018$) than the controls. In the subcohort study, marginal differences were seen in education level [$t(62.67) = 1.82$, $p = 0.074$] and gender [$\chi^2 (1) = 2.53$, $p = 0.111$], with slightly higher education level and proportion of women in the control group. In the ANCOVAs of outcome measures, age, gender, and living alone were included as covariates in the main cohort study and education level and gender as covariates in the subcohort study.

The choir singers and control subjects were well-matched for the level of non-musical cognitive and physical activities. In the main cohort study, the three groups differed highly significantly in the frequency of group singing at old age [age $\geq 60$ years; $F(2,154) = 209.34$, $p<0.001$], middle age [age 30–59 years; $F(2,154) = 56.84$, $p<0.001$], and young age [age 13–29 years; $F(2,154) = 32.81$, $p<0.001$]. Post hoc tests showed that both the high activity and low activity choir singing groups reported more frequent group singing than the control group at each life era ($p<0.005$ in all) but did not differ from each other in group singing at old age ($p = 0.948$). The high activity choir singers had more frequent group singing than the low activity choir singers at middle ($p<0.001$) and young ($p<0.001$) ages, confirming that the low activity choir singers had indeed started their choir hobby mostly after the age of 60 years. In the subcohort study, the choir singers had clearly more frequent group singing activity than the control subjects at each life era ($p<0.001$ in all).

**Table 2. Demographic characteristics of the participants.**

| | Questionnaire study | | | | Neuropsychological study | | |
|---|---|---|---|---|---|---|---|
| | High activity choir singers (N = 48) | Low activity choir singers (N = 58) | Control subjects (N = 35) | p value | Choir singers (N = 39) | Control subjects (N = 35) | p value |
| Age (years) | 72.8 (5.7) | 69.2 (4.5) | 70.2 (6.5) | 0.004 (F) | 70.9 (5.9) | 69.7 (6.6) | 0.396 (t) |
| Gender (women / men) | 38 / 10 | 38 / 20 | 48 / 8 | 0.034 ($\chi^2$) | 26 / 13 | 29 / 6 | 0.111 ($\chi^2$) |
| Education level[a] | 4.3 (2.2) | 4.2 (1.9) | 4.7 (2.1) | 0.375 (F) | 3.8 (1.7) | 4.7 (2.3) | 0.074 (t) |
| Living situation (alone / together) | 20 / 27 | 16 / 41 | 27 / 27 | 0.056 ($\chi^2$) | 15 / 24 | 17 / 18 | 0.381 ($\chi^2$) |
| Cognitive activity (old adulthood)[b] | 13.2 (3.0) | 12.7 (2.6) | 12.4 (3.7) | 0.446 (F) | 12.5 (2.4) | 12.6 (3.9) | 0.857 (t) |
| Physical activity (old adulthood)[c] | 9.1 (2.4) | 9.7 (2.8) | 9.3 (3.1) | 0.558 (F) | 9.7 (2.4) | 9.8 (3.4) | 0.895 (t) |
| Group singing (old adulthood)[d] | 3.9 (0.3) | 4.0 (0.4) | 0.9 (1.4) | <0.001 | 4.0 (0.5) | 0.6 (1.1) | <0.001 |
| Group singing (middle age)[d] | 3.6 (1.0) | 1.6 (1.5) | 0.8 (1.4) | <0.001 | 2.6 (1.6) | 0.9 (1.5) | <0.001 |
| Group singing (young adulthood)[d] | 3.3 (1.3) | 1.9 (1.5) | 1.0 (1.4) | <0.001 | 2.6 (1.6) | 0.8 (1.3) | <0.001 |
| Group singing years total | 27.3(14.2) | 6.3 (2.8) | | <0.001 | 15.8(14.3) | | |

Data are mean (SD) unless otherwise reported. Abbreviations: F = one-way ANOVA, $\chi^2$ = chi-square test, t = independent-samples t test.

[a]Education level according to the Unesco International Standard Classification of Education: range 1 (primary education) to 8 (doctoral level).

[b]Cognitive activity level during older adulthood based on the Lifetime of Experiences Questionnaire (LEQ) scores.

[c]Physical activity level during older adulthood based on the Lifetime of Experiences Questionnaire (LEQ) scores.

[d]Self-rating of group singing activity on a 6-point Likert scale ranging from 0 (never) to 5 (daily).

## Neuropsychological test results

The neuropsychological test scores of the choir singer and control groups are shown in Table 3. As illustrated in Fig 1A, the choir singers were significantly better than the controls on the verbal flexibility subdomain of executive functions [F(1,64) = 4.09, p = 0.047]. There were

**Table 3. Group differences in neuropsychological tests.**

| Questionnaire | Choir singers (N = 39) | Control subjects (N = 35) | p value[a] |
|---|---|---|---|
| General cognition | 25.9 (2.7) | 25.7 (2.9) | 0.275 |
| Executive function: Verbal Flexibility | 17.3 (5.7) | 16.0 (5.2) | **0.047** |
| Executive function: Shifting | 37.3 (26.6) | 36.9 (23.8) | 0.848 |
| Executive function: Inhibition | 101.2 (44.1) | 98.0 (47.1) | 0.912 |
| Processing speed | 70.6 (16.3) | 75.3 (16.1) | 0.637 |
| Working memory | 34.03 (4.8) | 33.51 (5.6) | 0.469 |
| Arithmetic | 13.24 (3.3) | 13.37 (3.3) | 0.619 |
| Episodic memory: Immediate recall | 41.0 (9.7) | 42.3 (8.5) | 0.961 |
| Episodic memory: Delayed recall | 16.3 (6.7) | 16.4 (6.2) | 0.646 |
| Verbal skills | 60.1 (11.8) | 59.5 (12.9) | 0.151 |

Data shown as mean (SD).

[a]The p value is the Group effect from univariate ANCOVA with education level and female gender as covariates.

Significant effects shown in bold font.

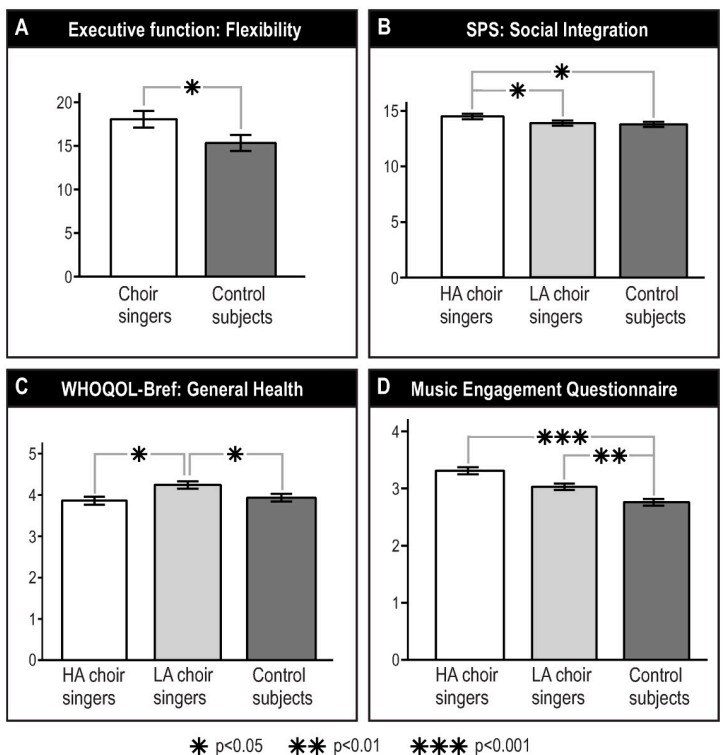

**Fig 1. Significant group differences between choir singers and control subjects in (A) verbal flexibility, (B) social integration, (C) general health, and (D) music engagement.** Bars show data as mean (SEM), with the significance level of the pair-wise group differences indicated with asterisks. Abbreviations: HA = high activity, LA = low activity.

no significant group differences in the other cognitive domains. For those tests which are standardized and have adequate Finnish normative data, and for which a cut-off value for impaired performance can be established (visual search, coding, digit span, arithmetic, word lists, vocabulary), there were no relevant differences between the choir singer and control groups on the proportion of participants scoring in the impaired range (see S1 Table, online only).

## Questionnaire results

The questionnaire scores of two choir singer groups and the control group are shown in Table 4. Significant group differences were observed in three questionnaires: the Social Integration scale of SPS [$F_{(2,151)}$ = 3.15, p = 0.046; Fig 1B], the General Health item of WHOQOL-Bref [$F_{(2,151)}$ = 3.61, p = 0.029; Fig 1C], and the MusEQ total score [$F_{(2,150)}$ = 8.53, p<0.001; Fig 1D]. There were no other significant effects. Post hoc tests indicated a different pattern of pair-wise group differences in the three questionnaires. In SPS Social Integration, the scores were higher in the high activity choir singers than in the low activity choir singers (p = 0.037) and control subjects (p = 0.023) whereas the low activity choir singers and control subjects did not differ (p = 0.850). In contrast, the WHOQOL-Bref General Health scores were higher in the low activity choir singers than in the high activity choir singers (p = 0.014) and control subjects (p = 0.037) but did not differ between the high activity choir singers and control subjects (p = 0.637). Finally, the MusEQ scores were higher in both the high activity and low activity choir singers compared to the control subjects (p<0.001 and p = 0.009, respectively), but did not differ between the two choir singer groups (p = 0.136).

**Table 4. Group differences in questionnaires.**

| Questionnaire | High activity choir singers (N = 48) | Low activity choir singers (N = 58) | Control subjects (N = 56) | p value[a] |
|---|---|---|---|---|
| CFQ total score | 24.3 (12.9) | 22.0 (11.3) | 22.6 (11.8) | 0.592 |
| PRMQ total score | 32.0 (8.4) | 30.3 (7.8) | 31.3 (7.6) | 0.690 |
| CES-D total score | 10.8 (6.0) | 9.1 (5.1) | 9.5 (5.7) | 0.332 |
| SPS total score | 84.9 (6.1) | 84.3 (6.4) | 83.9 (7.4) | 0.637 |
| SPS: Attachment scale | 14.6 (1.4) | 14.6 (1.5) | 14.1 (2.1) | 0.396 |
| SPS: Social Integration scale | 14.5 (1.2) | 13.9 (1.5) | 13.8 (1.7) | **0.046** |
| SPS: Reassurance of Worth scale | 13.1 (1.7) | 13.3 (1.5) | 13.4 (1.7) | 0.776 |
| SPS: Reliable Alliance scale | 14.6 (1.3) | 14.6 (1.6) | 14.8 (1.4) | 0.772 |
| SPS: Guidance scale | 14.9 (1.3) | 14.8 (1.4) | 14.6 (1.7) | 0.419 |
| SPS: Opportunity of Nurturance scale | 13.2 (1.9) | 13.2 (1.6) | 13.3 (1.7) | 0.779 |
| WHOQOL-Bref: Overall QOL | 4.1 (0.6) | 4.2 (0.5) | 4.3 (0.6) | 0.363 |
| WHOQOL-Bref: General Health | 3.8 (0.9) | 4.2 (0.6) | 4.0 (0.8) | **0.029** |
| WHOQOL-Bref: Physical QOL | 16.0 (2.7) | 17.0 (2.0) | 16.6 (2.5) | 0.133 |
| WHOQOL-Bref: Psychological QOL | 15.7 (2.0) | 16.0 (1.8) | 15.8 (2.0) | 0.918 |
| WHOQOL-Bref: Social QOL | 15.8 (2.5) | 16.1 (2.4) | 15.6 (2.7) | 0.586 |
| WHOQOL-Bref: Environmental QOL | 16.6 (2.0) | 17.1 (1.7) | 17.3 (1.5) | 0.135 |
| MusEQ total score | 3.2 (0.5) | 3.1 (0.4) | 2.8 (0.6) | **<0.001** |

Data shown as mean (SD).

[a]The p value is the Group effect from univariate ANCOVA with age, female gender and living alone as covariates. Significant effects shown in bold font.

## Discussion

In this study, our aim was to explore whether active participation in choir singing was associated with cognitive, emotional, and social well-being and QOL in healthy older adults. We used a comprehensive set of questionnaires and neuropsychological tests to assess participants' well-being and cognitive performance in a relatively large sample of older adults. Our hypotheses were that choir singers would have better performance in neuropsychological tests, especially in those measuring executive functions, and that they would experience better emotional, cognitive, and social well-being, compared to controls. These hypotheses were partially supported by the results, which showed that compared to the control group the choir singers had better verbal flexibility and they also experienced better social integration and health and had higher musical engagement in daily life.

In the subcohort study, the main finding was enhanced executive function in the domain of verbal flexibility in the choir singers compared to control subjects. This may reflect the specific verbal-cognitive demands of choir singing, especially the vocal production of lyrics while focusing concurrently on musical structure (the melody and rhythm of the song), auditory perception (perception of own voice and voices of other singers) and action (correcting own voice and adjusting it to others), musical instruction (following conductor's gestures, anticipating the next words), and emotional expression, which requires flexibility. The result is also in line with the findings of Fu et al. [33] that a 12-week group singing program had a positive effect on phonemic fluency in older adults. Also Mansens et al. [25] reported that music making, which in their study included both singing and playing a music instrument, was associated with better phonemic fluency. Similarly, Hanna-Pladdy and Gajewski [23] found that instrumental musical practice was associated with better phonemic fluency, in addition to better performance also in other tasks measuring cognitive flexibility. Together, these findings suggest that cognitive flexibility seems to be a core function, which is positively affected by musical

training and music activity in general during aging. At the neural level, a potential neurobiological mechanism for this effect could be striatal dopamine synthesis, which is known to influence the tuning of networks underlying the preservation of cognitive flexibility in aging [53] and which is increased through the reward and pleasure received from music [15] and during verbal learning [54].

Aside from verbal flexibility, we did not observe benefits of choir singing on any other cognitive domain, either in the neuropsychological tests or in cognitive self-report questionnaires (CFQ, PRMQ). Also in previous studies, findings regarding the effects of choir singing in other cognitive domains have been somewhat mixed. Some studies have reported that choir singing or singing in general is associated with better verbal skills, learning, and memory [25, 33] while others have not. In their randomized controlled trial (RCT), Johnson et al. [38] did not find any cognitive benefits from a community choir intervention for cognitively healthy older adults. Overall, the extent to which choir singing can have positive far transfer effects in the more general domains of executive function, attention, and memory function, still remains unknown.

In the main cohort study, we found that choir singers experienced better general health and social integration compared to the control group. Overall, this is in line with previous studies of older adults reporting that choir singing is associated with reduced loneliness, improved QOL, physical health, and interest in life, and increased general activity [36–39]. Interestingly, in our study, these effects seemed to be mediated by the length of the choir singing activity: better social integration was seen only in the high activity choir singers who had more than 10 years of choir experience whereas better general health was seen only in the low activity choir singers who had started choir singing at older age and had ≤10 years choir experience. As the two choir singer groups were well-matched with regards to how often they currently participated in choir singing as well as in non-musical (cognitive and physical) activities and as the demographic variables in which the groups differed (age, gender, living alone) were controlled for in the analysis, it is yet unclear why the two choir singer groups showed different effects. A potential explanation could be that for the high activity choir singer group, the long-standing choir activity and the personal relationships formed with the other choir members have become an integral part of their social life—a hobby that unites them and keeps them socially connected. In turn, for the low activity choir singers, participation in the choir activity may be more motivated by an aim to maintain better health during aging, as a part of a healthy and active lifestyle.

Our study has some methodological limitations, which should be taken into account when evaluating its findings. First of all, the choir arm of the study is based on a natural sample of older adults participating in choir singing, and therefore we have no experimental control of the choir training, for example in terms of quality and quantity of training, skill level, and goals and motivation, and how these factors could impact the results. On the other hand, this same facet makes the study more representative of real life and provides ecological validity, although the relatively small sample size (especially of the subcohort study) somewhat limits the generalizability of the findings. In recruiting the participants, we made a clear distinction between actual choir singing and other types of group singing (e.g., taking part in singalong sessions, singing as an audience member at church, concerts or sports events etc.), as the goal-oriented and learning-related cognitive component applies more to choir singing. Second, given that this was a cross-sectional study, the results are largely correlational and no conclusions can be made about the causality of findings and the long-term effects of choir singing on neurocognitive aging; for this, more large-scale studies with randomized controlled trial (RCT) and longitudinal cohort designs are needed. The follow-up data from our cohort (study ongoing) may be able to shed more light on this issue. In future studies, also an active control

condition (another specific hobby) would be needed. Third, this study focused on cognitive, emotional, and social outcomes and therefore it is not able to determine the potential physiological (e.g., cardiorespiratory), motor (e.g., balance, posture), hormonal (e.g., oxytocin, cortisol), and neural (brain structure and function) effects of choir singing, which would be needed to establish a more comprehensive picture of the role of choir singing in healthy aging.

## Implications

The results of the present cross-sectional study suggest that regular participation in choir singing as a hobby is associated with better verbal flexibility in healthy older adults, but positive effects on other cognitive functions are limited. In addition, long-standing choir participation was linked to better social integration whereas choir activity, which was started at older age, was linked to better general health. Together with previous studies, there is emerging evidence that singing in a choir may provide an accessible and useful way to stave off the negative social-emotional sequelae (e.g., loneliness, social isolation) and cognitive decline, which are typically associated with aging. This is highly important to society because the onset and progress of age-related cognitive decline and mood disorders are known to be closely linked to reduced social interaction [6, 7, 55] and executive dysfunction [2, 3, 8], and there is urgent need for different lifestyle interventions that can be utilized to support healthy aging [5].

## Supporting information

**S1 Table. Proportion of participants scoring in the impaired range in standardized tests with Finnish normative data.**
(DOCX)

## Acknowledgments

We warmly thank the participants of this study for their time and commitment, Dr. Mari Tervaniemi for her advice and support in planning the study, Dr. Jussi Virkkala and MSc Tommi Makkonen for their technical assistance in data collection.

## Author Contributions

**Formal analysis:** Emmi Pentikäinen.

**Methodology:** Teemu Paajanen.

**Supervision:** Teppo Särkämö.

**Writing – original draft:** Emmi Pentikäinen.

**Writing – review & editing:** Emmi Pentikäinen, Anni Pitkäniemi, Sini-Tuuli Siponkoski, Maarit Jansson, Jukka Louhivuori, Julene K. Johnson, Teemu Paajanen, Teppo Särkämö.

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
