## [Decision Letter · Decision Letter 0]

24 Nov 2020

PONE-D-20-19456

Beneficial effects of choir singing on cognition and well-being of older adults

PLOS ONE

Dear Dr. Pentikäinen,

Thank you for submitting your manuscript to PLOS ONE. After careful consideration, we feel that it has merit but does not fully meet PLOS ONE’s publication criteria as it currently stands. Therefore, we invite you to submit a revised version of the manuscript that addresses the points raised during the review process.

We look forward to receiving your revised manuscript.

Kind regards,

Laura Zamarian

Academic Editor

PLOS ONE

Journal Requirements:

Reviewers' comments:

Reviewer's Responses to Questions

**Comments to the Author**

1. Is the manuscript technically sound, and do the data support the conclusions?

Reviewer #1: Yes

Reviewer #2: Yes

2. Has the statistical analysis been performed appropriately and rigorously? 

Reviewer #1: Yes

Reviewer #2: Yes

3. Have the authors made all data underlying the findings in their manuscript fully available?

Reviewer #1: Yes

Reviewer #2: Yes

4. Is the manuscript presented in an intelligible fashion and written in standard English?

Reviewer #1: Yes

Reviewer #2: Yes

5. Review Comments to the Author

Reviewer #1: This is an interesting and well written article describing the results of a cross-sectional study examining the effects of choir singing in cognition and wellbeing of older adults.

Overall the paper is clearly laid out and all sections are articulated well. There are only a few minor things for consideration that I have outlined below.

Please consider including a reference to the study design in the title, ie. cross-sectional study.

By “long-time” and “short-time” choir experience do you mean “long-term” and “short-term”?

Introduction is clear and succinct and supported by appropriate current references.

Suggest writing “executive function” out in full rather than abbreviate to EF.

A clear summary of current published knowledge in the area and justification for the study.

Method and Results are clearly expressed. Appropriate use of tables and figures.

Discussion and Implications are comprehensive and appropriate reflect on findings and study limitations in relation to previous research.

Possible typo on p12 line 359 “positive effects on other cognitive functions…”

Reviewer #2: Review - Beneficial effects of choir singing on cognition and well-being of older adults

This is a very interesting and well-written manuscript.

I have only few comments:

The dichotomy between long-time (>10 years, N = 48) and short time (≤10 years, N = 58) choir singers seems arbitrary and is not well explained. In fact, the differences in the results are not easily explainable and remain speculative. I miss a minimum criterium of choir singing (e.g., include only participants who have at least two or three years of choir singing). In order to find beneficial effects, choir singing has to extend over a certain time span. Then I would enter the duration of choir singing as a continuous variable instead as a categorical and see the effects.

I would not enter arithmetic as a working memory test. Though arithmetic relies on working memory, it requires also other cognitive domains (e.g., procedural knowledge, conceptual knowledge). Please list arithmetic as a separate domain.

Table 3: Are these values in the average of standardized norms? Above/below? Are there differences between groups (proportion of participants scoring in the impaired range)?

6. PLOS authors have the option to publish the peer review history of their article (what does this mean?). If published, this will include your full peer review and any attached files.

Reviewer #1: **Yes: **Dr Jeanette Tamplin

Reviewer #2: No

---

## [Author Response · Author response to Decision Letter 0]

17 Dec 2020

We would like to thank the Reviewers for their valuable comments, which have greatly helped us to improve the manuscript. Below, we have listed our responses to each point made by the Reviewers. 

Reviewer 1

1. Please consider including a reference to the study design in the title, i.e. cross-sectional study.

Response: We agree that adding the study type to the title is a good idea and can avoid potential misunderstanding. We have now changed the title to “Beneficial effects of choir singing on cognition and well-being of older adults: a cross-sectional study”

2. By “long-time” and “short-time” choir experience do you mean “long-term” and “short-term”?

Response: Thank you for noting this. In hindsight, we agree that the terms “long-time” and “short-time” were somewhat awkward and potentially misleading. The suggested terms “long-term” and “short-term” are better, but in our view, “short-term” refers to a time period shorter than the one we mean here (choir singing experience ranging from 1 to 10 years); also the Merriam-Webster dictionary defines “short-term” as something “occurring over or involving a relatively short period of time”, often less than 6 months / 1 year. In revising the terms, we opted to use the terminology used previously by Hanna-Pladdy and MacKay (2011, Neuropsychology) who divided their older adult musicians to low activity (1–9 years) and high activity (>10 years) groups based on years of musical experience throughout their life span. Thus, we have now changed “long-term” and “short-term” to “low activity” and “high activity”, respectively, across the manuscript.

3. Introduction is clear and succinct and supported by appropriate current references. Suggest writing “executive function” out in full rather than abbreviate to EF. A clear summary of current published knowledge in the area and justification for the study.

Response: Thank you for the positive feedback. We have now replaced the EF abbreviation with “executive function” across the manuscript.

4. Discussion and Implications are comprehensive and appropriate reflect on findings and study limitations in relation to previous research. Possible typo on p12 line 359 “positive effects on other cognitive functions…”

Response: There was indeed a typo, which we have now corrected.

Reviewer 2

1. The dichotomy between long-time (>10 years, N = 48) and short time (≤10 years, N = 58) choir singers seems arbitrary and is not well explained. In fact, the differences in the results are not easily explainable and remain speculative. I miss a minimum criterium of choir singing (e.g., include only participants who have at least two or three years of choir singing). In order to find beneficial effects, choir singing has to extend over a certain time span. Then I would enter the duration of choir singing as a continuous variable instead as a categorical and see the effects.

Response: Our decision to use dichotomous classification was motivated by (i) previous studies utilizing the same approach in musicians and their findings that the duration of musical activity has an impact on the cognitive benefits incurred by music, (ii) the distribution of years of choir singing in our sample, and (iii) practical considerations of equal group sizes for statistical analysis.

i. Previously, in their study of healthy older adults, Hanna-Pladdy and MacKay (2011, Neuropsychology) divided the musician sample to low (1–9 years) and high (>10 years) activity group based on years of musical experience throughout the life span. They showed that compared to non-musician controls, only the high activity musicians had better performance in cognitive tests. Also other studies have shown that musical experience acquired earlier in life vs. more recently can have differential effects on cognitive functioning (Hanna-Pladdy & Gajeweski, 2012; Strong & Mast, 2018). In light of the comment of Reviewer 1, we have now adopted the terms used by Hanna-Pladdy and MacKay (2011) and replaced “short-time” with “low activity” and “long-time” with “high activity”, and cite this study also in the Methods section.

ii. In our sample, years of choir singing did not follow a normal distribution (see figure below), which is problematic for its inclusion as a parametric (continuous) variable in the analyses. The median value was 10 years, which was chosen as a natural cut-off point for the group classification. In the low activity choir singers, the mean of years of singing was 6.3 years and SD 2.8 years. In the high activity singers, these were 27.3 years and 14.2 years, respectively. We have added this information in the manuscript (see Table 2).

iii. Given that in the main cohort, the sample sizes of the controls (N=56) and choir singers (N=106) were unbalanced, which leads to unequal variances between samples that violates the assumption of equal variances in ANOVA and affects statistical power and Type I error rates (e.g., Rusticus & Lovato, 2014, Practical Assessment, Research & Evaluation). Including years of choir singing as a classifying variable results in relatively balanced group sizes (controls: N=56, low activity singers: N=58, high activity singers: N=48), differing only by 4%–11% from the optimal group size (162/3=54), as opposed to the 31% difference from optimal if two groups were used (N=56 and N=106; 162/2=81). Thus, our approach avoids the unequal sample size problem.

Regarding the minimum amount of choir singing, we actually included also a “minimum 1 year” criterion when recruiting the choir singers. Thus, all the choir singers had to be (1) currently singing in a choir and (2) having sung in a choir for at least one year. Unfortunately, this important information was omitted from the original manuscript by mistake; it has now been added to the Methods section.

2. I would not enter arithmetic as a working memory test. Though arithmetic relies on working memory, it requires also other cognitive domains (e.g., procedural knowledge, conceptual knowledge). Please list arithmetic as a separate domain.

Response: We agree that including arithmetic in the working memory domain is somewhat misleading (it was originally included there as it is belongs to the Working memory index of WAIS-IV, but since we are also using the visual span here, which is not from WAIS-IV, there is no need to adhere to the WAIS-IV guideline). We have now included the WAIS-IV Arithmetic score as a separate domain of Problem solving and reanalyzed the data for the Working memory (now WAIS-IV Digit span and FAT Visual span subtest) and Problem solving domains (WAIS-IV Arithmetic). This does not change the results (neither domain differs between the choir singers and controls).

3. Table 3: Are these values in the average of standardized norms? Above/below? Are there differences between groups (proportion of participants scoring in the impaired range)?

Response: Originally, we chose to report the raw scores (or their sums for the cognitive domains) of the neuropsychological tests because not all tests were standardized and have adequate Finnish normative data. For those tests which are standardized and which have adequate normative data, and for which a cut-off value for impaired performance can be established (visual search, coding, digit span, arithmetic, word lists, vocabulary), there were no differences between the choir singer and control groups on the proportion of participants scoring in the impaired range (see table below). Even though a participant may have scored in the impaired range in one or more of the tests, they did not have any diagnosed neurodegenerative diseases, which was our inclusion criterion.

Domain Measure % of subjects in impaired range Difference between groups (X2)

 Choir (N=39) Control (N=35) 

Processing speed WAIS-IV Visual search 17.9 17.1 0.93

 WAIS-IV Coding 10.3 0 0.05

Working memory WAIS-IV Digit span 5.1 8.6 0.56

Problem solving WAIS-IV Arithmetic 23.1 22.9 0.92

EM:

Immediate WMS-III Word lists 20.5 11.4 0.34

EM:

Delayed WMS-III Word lists 17.9 17.1 0.93

Verbal skills WAIS-IV Vocabulary 7.7 14.3 0.36

Abbreviations: EF=executive function, EM=episodic memory

---

## [Editor Report · Decision Letter 1]

23 Dec 2020

PONE-D-20-19456R1

Beneficial effects of choir singing on cognition and well-being of older adults: evidence from a cross-sectional study

PLOS ONE

Dear Dr. Pentikäinen,

I have read myself your paper and am very satisfied with the apported changes.

Before making a final decision, I would be very grateful, if you could add as supplemental material the table with the distributions of impairments in cognitive tests you are given in your answer-to-reviewers. In the text - results section, please also comment that "For those tests which are standardized and have adequate Finnish normative data, and for which a cut-off value for impaired performance can be established (visual search, coding, digit span, arithmetic, word lists, vocabulary), there were no relevant differences between the choir singer and control groups on the proportion of participants scoring in the impaired range (see table in the Supplemental material, online only)."

Kind regards,

Laura Zamarian

------------------

We look forward to receiving your revised manuscript.

Kind regards,

Laura Zamarian

Academic Editor

PLOS ONE

---

## [Author Response · Author response to Decision Letter 1]

5 Jan 2021

We have now added the supplemental material, as well as a comment in the results section referring to the table in the supplemental material.

---

## [Editor Report · Decision Letter 2]

6 Jan 2021

Beneficial effects of choir singing on cognition and well-being of older adults: evidence from a cross-sectional study

PONE-D-20-19456R2

Dear Dr. Pentikäinen,

We’re pleased to inform you that your manuscript has been judged scientifically suitable for publication and will be formally accepted for publication once it meets all outstanding technical requirements.

Kind regards,

Laura Zamarian

Academic Editor

PLOS ONE
---

## [Editor Report · Acceptance letter]

11 Jan 2021

PONE-D-20-19456R2 

Beneficial effects of choir singing on cognition and well-being of older adults: evidence from a cross-sectional study 

Dear Dr. Pentikäinen:

I'm pleased to inform you that your manuscript has been deemed suitable for publication in PLOS ONE. Congratulations! Your manuscript is now with our production department. 

Kind regards, 

on behalf of

Dr. Laura Zamarian 

Academic Editor

PLOS ONE